# Roadmapping 5.0 Technologies in Agriculture: A Technological Proposal for Developing the Coffee Plant Centered on Indigenous Producers’ Requirements from Mexico, via Knowledge Management

**DOI:** 10.3390/plants11111502

**Published:** 2022-06-03

**Authors:** David Israel Contreras-Medina, Sergio Ernesto Medina-Cuéllar, Juan Manuel Rodríguez-García

**Affiliations:** Departamento de Arte y Empresa, División de Ingenierías Campus Irapuato-Salamanca DICIS, Universidad de Guanajuato, Carr. Salamanca-Valle de Santiago km 3.5 + 1.8, Comunidad de Palo Blanco, Salamanca 36885, Mexico; se.medina@ugto.mx (S.E.M.-C.); juanrodriguez@ugto.mx (J.M.R.-G.)

**Keywords:** coffee plant, indigenous small producers, Industry 5.0 technologies, roadmapping, knowledge management

## Abstract

The coffee plant, with more than 40 billion shrubs, 9 million tons of grains produced, and 80% of its production accounted for by small-scale producers, has been severely damaged since the emergence of *Hemileia vastatrix* and *Hypothenemus hampei*. Despite technological support, these pests have caused 20% to 40% production losses, a 50% to 60% deficit in performance, and a cost of between USD 70 million and USD 220 million to the world economies, which forces us to rethink actions centered on people as the key elements to develop appropriate solutions. For this, the present study presents a technological proposal centered on small indigenous coffee producer requirements for introducing Industry 5.0 technologies, considering roadmapping, knowledge management, statistical analysis, and the social, productive, and digital contexts of five localities in Mexico. The results show a correlation between monitoring and control, soil analysis, the creation of organic fertilizers, accompaniment, and coffee experimentation, as the actions to be implemented, proposing the introduction of a mobile application; sensors, virtual platforms, dome-shaped greenhouses, and spectrophotometric technology as relevant technologies centered on indigenous coffee producers’ requirements. This study is important for policymakers, academics, and producers who wish to develop strategies centered on people in Mexico and the world.

## 1. Introduction

Historically, plants have played an essential role in the evolution of human beings. Due to their role in the production of oxygen, humidity regulation, the stability of the climate and soil, fuel, food, and medicine for protecting the coronary arterial wall and against oxidative inflammatory disease [1,2,3], plants have become an essential element in the life and culture of human beings [4].

Worldwide, around 300,000 edible plants exist, of which approximately 6000 are cultivated by humans, using about 200 to produce food [5,6]. Among these plants is coffee, with more than 40 billion shrubs in the world, 9 million tons of grain produced, and almost 3 billion kilograms consumed during the period 2019–2020 [7], which is becoming one of the most demanded beverages in the world [8,9,10].

The coffee plant, divided into arabica (*Coffea arabica*) and canephora (*Coffea canephora*) as its two most important varieties, is a tropical tree with green leaves and flowers that give an oval fruit called a coffee bean. Its development requires best management practices, temperatures between 15 and 30 degrees Celsius, pluvial precipitations of 1500 to 3000 mm for a proper development, and 9 to 11 months from flowering, on average, to produce a ripe cherry [11]. However, the sensitivity of the plant in relation to poor management practices and climatic change has favored the presence of pests [12] such as coffee leaf rust (*Hemileia vastatrix*) [13,14] and the coffee berry borer (*Hypothenemus hampei*) [15]. These pests have caused 20% to 40% worldwide agricultural production losses, a 50 to 60% deficit in performance, and a cost of around USD 70 million to USD 220 million to the world economies [16,17]. 

Due to the above, strategies have been proposed to combat pests using the support of technology. These actions are evident in the creation of manuals for sustainable coffee production [18]; in the development of resistant varieties for genetic improvement; in the increase in quality [19]; in the manufacture of organic fertilizers for treatment [20]; and in the use of drones to detect the presence of coffee leaf rust [21]. However, despite this, to date, the existence of coffee leaf rust has been devastating for various growing regions in the world [22,23], estimating 30% crop production losses in Latin America alone [24], representing 18.4% of the world, in 2020. Although these efforts are significant, little attention has been paid to people as a critical element in developing appropriate solutions [25], and, in the context of coffee, of which 80% worldwide is produced on a small scale [26], studies require the inclusion of producers located in Africa, Asia, and America [19,27,28].

Since the 20th century, emphasis has been placed on the inclusion of human beings in order to understand their needs, desires, and opportunities [29,30]. The European Union has promoted this line of connection with society with the program “Science with and for Society (SwafS)” to connect science and technological advances [31]. The program of people-centered employment support from the Organization for Economic Co-operation and Development (OECD) understands the individual circumstances to connect persons with good jobs [32], or, with Industry 5.0, to place innovations and technologies under a human-centered approach [33,34]. 

With the arrival of Industry 5.0 [35], the human being is the pivot for interacting with technologies [36]. However, the introduction and use of digital tools benefit less than 50% of the worldwide population, despite the fact we are living in the digital age, with advances and significant transformations for society [37,38,39]. 

The evolution of Industry 5.0 has seen the application of various technologies. For example, the migration of agriculture towards its industrialization using fossil fuels in Industry 1.0 increased production. Other examples include the mechanical devices of Industry 2.0, the electronic tools of Industry 3.0, the Internet of Things of Industry 4.0, and the robots and intelligent systems of Industry 5.0. However, introducing these technological tools to the primary stage of food production systems has been complex for peasants and indigenous people who have managed their crops by hand or with animals’ assistance for centuries [36,40]. Because the concept of Industry 5.0 is human-centered [41], it is relevant to understand these groups’ specific requirements and preserve their traditional knowledge by directing digital technologies to their communities [42,43,44]. 

Globally, people dedicated to coffee cultivation have different situations, capacities, and needs. For example, in Africa, where the average age of farmers is 60 years, poverty is higher than in other producing regions, with marginal resources in technology and limited financial capital, reducing the attention towards efficiency, cost reduction, and sustainable practices [45]. Asia, particularly India, is in an extreme poverty situation, with an average age of 38 years for its producers [46], where resources are used for developing biological and socio-economic surveys for disease treatment and sustainable production [47]. In America, specifically in Mexico, local producers live in poverty conditions, with an age range between 36 and 60 years, where government resources are used for the renewal of plants and to combat the presence of rust [48]. This heterogeneity of contexts in today’s digital age makes it pertinent to build specific strategies within the various regions using technology.

Currently, emphasis is beginning to be placed on studies related to technologies centered on human beings in different industries. For example, in the aviation context, the author of [49] reported on human and technology cooperation, how people function with transportation systems, their roles, management practices, and problems of the combination of individuals with automated devices. In tourism, ref. [50] showed the dependence of this industry on technologies by creating new physical and virtual objects, mentioning an overlap with reality, moving away from human-centered design. In healthcare, ref. [51] presented a review of technologies as simulation systems for training, concluding with the limited consideration of the human-centered approach and the emphasis in the sector to create digital tools from technical elements. The author of [52] presented a study for proposing tools and methods for human–machine interaction, registering the human and technology interaction as an emergent field of investigation. 

In this line, to understand the requirements of people and developing technologies in agriculture, ref. [53] conducted a study centered on people to explore their ideas and designs to make weeding more manageable, such as a lighter stirrup weeder, rubber material in bicycle handlebar grips, forked jembes, or reused machetes, according to farmers’ needs in Kenya. In [54], a project called “SmartAgro”, involving students and their fathers, was created to attend to climate change and droughts by optimizing the level of water applied to coffee plants using the Internet of Things and data analysis, which increased the number of beans and the size of beans. In [55], small producers developing organic fertilizers were studied, based on the need to reduce production costs and contribute to caring for the coffee-growing soil in El Salvador. Moreover, [56], exploring coffee farmers, introduced new technologies in roasting to add value and boost consumption and entrepreneurship according to the needs for improving production and quality in India. 

In Mexico, similar studies are as follows. The author of [57], through a user-centered methodology involving stakeholders, developed knowledge by creating and refining technologies such as farming practices and hubs for improving maize and wheat production at MasAgro. Today, more than 40% of the participants have adopted at least one innovation. The authors of [58] reported on Transformation Laboratories (T-Labs) as a human-centered participatory space, combining technologies and methods to explore chinampa agriculture degradation through the needs of agricultural producers to promote dialog. The study of [59] explored coffee producers’ needs and perspectives using the roadmapping method, resulting in the proposal to install a semi-open greenhouse adapting sensor technologies and a WIFI module to increase production in the indigenous region of Guerrero, Mexico. The authors of [60] studied mezcal producers and proposed the use of an easy-to-use mobile application and a mezcal tech-hub as technological opportunities to improve the dynamics and interaction with external agents in Oaxaca, Mexico. In this line, an additional background is presented in Table 1.

The previous observations clarify that technologies vary across countries, sectors, and people. These variations are evidenced in the results issued by the World Bank, which, through the Digital Adoption Index, shows the differences in digital adoption among governments, businesses, and people. For example, Mexico has a value of 0.60 (on a scale of 0–1), registering a gap of 0.27 from the best country, Singapore, with its lowest result among people at 0.43, a difference of 0.48 compared with the best country, Hong Kong [68], evidencing the low adoption of technologies by the Mexican population. This result could be the reason why the Mexican government started its efforts to develop the Industry 4.0 model centered on human beings, especially for the attention of farmers in areas of high and very high marginalization [69,70].

Studies that involve interaction with humans have presented several methodologies to identify the technological requirements of stakeholders. For example, [71] used a Bayesian network methodology to determine the relationship of variables in technology management in a financial sector company in Mexico. Another example is the technology acceptance model, considering the usefulness and ease of using technologies so that there is an intention towards using the same technologies, introducing the model in contexts such as health [72,73]. A further example is the method applied by [74], involving technology-based entrepreneurship opportunities through interviews; however, a flexible methodology that interacts with society to visualize a specific strategy is technology roadmapping. 

Proposed by [75], the technology roadmapping methodology involves the expectations, current situation, and strategies to visualize the actions through technologies. This method is an open method for knowledge management through personal interaction to explore technological requirements; moreover, it has already been applied and validated in the agricultural environment of Mexico, particularly that of coffee in the states of Guerrero and Chiapas [59,61].

Invariably, technologies and society are closely related. An adequate technological change must result from the interaction between local and formal knowledge, rules, and the implication of diversity adaptation [76] from the beginning, detecting human needs. To fill these gaps, this research presents a case study that aims to obtain information based on the coffee producer approach in the localities of El Pajarito, Llano Coyul, Ocotal, and Guadalupe in the municipality of Santiago Lachiguiri, in addition to Buenavista in San Juan Guichicovi, in the Isthmus of Tehuantepec in Oaxaca, Mexico. The objectives are as follows:To explore coffee’s social, productive, and digital contexts in Oaxaca, Mexico;To determine technological routes through the expectations, current situation, and actions that could be implemented by indigenous producers in coffee cultivation in Oaxaca, Mexico;To propose digital technologies based on actions to be implemented by indigenous coffee producers from Oaxaca, Mexico, correlating social and productive contexts.

This research contains five sections. The first presents the introduction explaining the importance of plants, the transcendence of coffee, its main varieties, and the impact of *Hemileia vastatrix* on production, development, and economy in the world. 

After this, the strategies developed to combat it and the absence of human beings’ participation in the development of solutions are highlighted. In this line, the historical role that human beings have played and its importance for economies and industry such as Industry 5.0. are recorded, pointing out the existing gap with the primary food production systems. In addition, this study presents the characteristics of different social contexts that surround coffee production. A literature review on technologies centered on human beings worldwide and in the agricultural context in Mexico is presented, ending with this study’s objectives. The second stage presents the materials and methods, registering the place of the survey, producers, and form of data collection, exploring (1) the productive and social contexts; (2) the expectations, current situation, and actions to be implemented by coffee producers; and (3) the technological proposal. After this, this study establishes the hypothesis and the analysis and validation of the information. In the third stage, the results are presented in a social, productive, and digital context, in addition to the process and technology routes, statistical analysis, and technological proposal, using Industry 5.0 technologies, while in the fourth stage, a discussion of the results is presented. The fifth stage presents the conclusions, limitations, and new study routes visualized from the results.

## 2. Materials and Methods

### 2.1. Information Gathering

The present study was carried out on 34 indigenous coffee producers from the localities of El Pajarito, Llano Coyul, Ocotal, Guadalupe, and Buenavista in the municipalities of Santiago Lachiguiri and San Juan Guichicovi in Oaxaca, Mexico. These municipalities account for 39.34% of the cultivated surface, equivalent to 55,548 ha, estimating a density of between 1600 and 2000 plants by hectare, and contributing to 42.88% of the production of the Isthmus region [12,77,78]. 

Initial contact was through the representatives of the local indigenous organization producing coffee in the Isthmus region, explaining the objective, the area of attention, and the evolution of this study in the southern Pacific region in Mexico. Accompanied by a technician, coffee producers were visited in their localities and consulted about if they wanted to participate in this study, being notified of the objective. The interaction was through face-to-face questions and field visits during the period of July to November 2018, collecting information, considering three stages:

First phase: For exploring social and productive contexts, information about age, gender, years spent producing coffee, family members, sale price, level of production, crop area, cultivated varieties, type of flora, pruning, frequency of pruning, type of fertilizer used, perception about the effectiveness of the fertilizer, application period, incidence of pests on plants, and additional crops was collected following the studies of [79,80,81,82], based on the assertion that social conditions must be considered for digital transformation within the rural context [83,84]. This stage was formalized through the following questions:Regarding social information, it was asked: “As a coffee producer, what is your age, gender, length of time producing coffee, and the number of family members?”In relation to the productive context, it was asked: “As a coffee producer, which is your level of production, varieties of coffee, cultivated area, type of flora, pruning, frequency of pruning, type of fertilizer used, perception about its effectiveness, application period, the incidence of pest on the plant, additional crops managed in the coffee-growing area, and sale price?”

The second phase: For identifying technological routes, the expectations, current situation, and actions to be implemented regarding coffee were determined according to the technology roadmapping methodology [75] using the following questions. It is essential to mention that these questions were constructed and validated based on previous studies [59,61].
Regarding expectations, it was asked: “As a coffee producer, how would you like the coffee crop to be in the region in the future?”Concerning the current situation, it was asked: “As a coffee producer, what is the current situation of the coffee crop in the region?”In relation to the strategies, it was asked: “As a coffee producer, which actions can be implemented in the coffee crop to meet the current situation’s expectations?”

Third phase: For the technological proposal, the information of the indigenous coffee producers was considered, as well as the availability of resources, following the line of [85] regarding the evaluation and selection of technology. In this sense, the requirements were assessed through statistical analysis following the studies of [86,87], based on the frequency of social and productive information, and actions to be implemented via technological routes, evaluating the normality of the distribution and applying parametric/non-parametric tests for checking statistical correlations, following [60,61,62]. With the calculation of statistical correlations, results were linked with the availability of resources in regard to Industry 5.0 technologies, following the line of [41], considering their implementation in similar contexts. Questions for this stage were as follows:In relation to statistical correlations, the question was: “Is there a statistical correlation between social and productive information and actions to be implemented by indigenous coffee producers?”Regarding the technological proposal, the question was: “Based on statistical correlations and the availability of resources, what type of technologies of Industry 5.0 can be introduced by indigenous coffee producers that have been used in similar contexts?”

As a part of this methodological model, an exploration of digital tools such as cell phones, computers, and internet availability in the localities of El Pajarito, Llano Coyul, Ocotal, Guadalupe, and Buenavista was added, in addition to the description of the coffee production process and materials used in it. 

Combining the three stages, the hypothesis to probe was as follows:

The technological proposal can improve the local development of the coffee tree due to its availability of resources and application in a similar context based on the requirements of indigenous coffee producers. 

To facilitate the responses of indigenous coffee producers, the medical consultation analogy was used to exemplify the activity under the expectation of a healthy human being, the current disease situation, and the action that must be developed to achieve the expectation from the current situation. The application of analogies is recommended by [88] and has also been applied successfully in studies in the same context, with indigenous and peasant coffee producers from Guerrero and Chiapas, Mexico [59,61].

### 2.2. Information Analysis and Validity

The results of the following were analyzed, registered, tabulated, and ordered according to their frequency of appearance from the most frequently chosen to the least frequently chosen, following [89]: expectations, current situation, and actions; social and productive information such as gender, length of time producing coffee, sale price, level of production, and coffee-growing area; and environmental data such as the variety of coffee, type of flora, pruning, frequency of pruning, type of fertilizer used, perception about its effectiveness, application period, the incidence of pests on plants, and additional crops managed in the coffee-growing area. Age and family members were grouped in intervals following the scheme of [90]. For this, NVIVO v11 software, IBM SPSS Statistics v21 for statistical analysis, and VISIO v 2016 for technological diagrams were used.

The identity of indigenous coffee producers was certified by the technician of the local indigenous organization producing coffee. 

## 3. Results

### 3.1. Social, Digital, and Productive Contexts of Indigenous Coffee Producers

The social, productive, and digital contexts of coffee producers from Oaxaca, Mexico, present different particularities within the localities of El Pajarito, Llano Coyul, Ocotal, Guadalupe, and Buenavista. 

El Pajarito has a population of 66 people and is located 1253 m above sea level (m.a.s.l), with indigenous and Spanish languages. The degree of schooling is 6.04 years for men and 5.36 years for women (primary and early high school); none of the dwellings have a computer, laptop, or tablet, 52.6% have a cell phone, and 52.6% have internet. For Llano Coyul, with 146 people, at 960 m.a.s.l., with indigenous and Spanish languages, the degree of schooling is 6.79 years for men and 5.03 years for women (primary and almost the first year of high school); a total of 4.5% of households have a computer, laptop, or tablet, 65.9% have a cell phone, and 59.09% have internet. Ocotal has 288 inhabitants and is 498 m.a.s.l., with indigenous and Spanish languages. The degree of schooling is 5.05 years for men and 4.36 years for women (primary school); a total of 3.3% of households have a computer, laptop, or tablet, 32.2% have a cell phone, and 6.6% have internet. Guadalupe has 555 people and is located at an altitude of 1022 m.a.s.l., with inhabitants speaking indigenous and Spanish languages. The degree of schooling is 6.77 years for men and 5.28 years for women (primary and early high school); a total of 8.7% of households have a computer, laptop, or tablet, 48.09% have a cell phone, and 29.5% have internet. Buenavista has 873 inhabitants and is located 1325 m.a.s.l., with indigenous and Spanish languages. The degree of schooling is 5.92 years for men and 4.36 years for women (primary school); a total of 2.4% of households have a computer, laptop, or tablet, 56.1% have a cell phone, and 1.05% have internet [91]. Regarding economic status, both municipalities have a range of between 19.4 and 22.4 people in extreme poverty [92] (see Figure 1 and Table 2).

In terms of the characteristics of coffee producers from the localities of El Pajarito, Llano Coyul, Ocotal, Guadalupe, and Buenavista, 91.1% are men and 8.8% are women; the average age is 63 years for males and 52 years for females; and the time spent producing coffee is 41.2 years for men and 11.5 years for women. Regarding family members, 3.6 persons on average are recorded (see Table 3). 

The production level per harvest is 285 kg (on average by ha). In terms of the varieties used by coffee producers, 47.2% use Catimor, 14.7% use Oro Azteca, 14.7% use Bourbon, 11.7% use Geisha, and 11.7% use others. In regard to the coffee tree age, 32.4% are less than two years old, 52.9% are two to seven years old, and 14.7% are more than seven years old. The cultivated area is 3.27 hectares (on average). The flora around the coffee cultivation area is 58.8% Chalum tree, 20.6% Cuil, and 20.6% other. Regarding the pruning of coffee trees, 61.7% do not prune their coffee trees, while 38.3% do; the frequency of pruning is 20.7% every year, 5.8% every two years, 5.8% every three years, and 5.8% every four years, while 61.9% of the respondents do not prune their trees. Concerning the fertilizers/pesticides used against rust, 52.9% use oxicloruro, 11.7% use organic products, 14.8% use others, and 20.6% do not use fertilizers. In the application of compost to the soil, 20.6% do not apply compost to the soil, while 79.4% do. Regarding the type of compost, 38.2% is organic, 26.4% is compost, and 14.7% is coffee shells, while 20.7% of the respondents do not use soil compost. In terms of additional crops, none are present in 47.1% of the cases, and maize and beans are present in 52.9% of the cases. The sale price is USD 1.94 per kilogram (MXN 40.33). The perception of pest incidence on coffee crops is 61.8% nothing, 26.5% little, and 11.7% a lot (see Table 4).

The process of coffee production in the localities of El Pajarito, Llano Coyul, Ocotal, Guadalupe, and Buenavista involves the following activities: (1) the local organization provides the seed to the producer; (2) the seed is planted in a hotbed; (3) the hotbed is transferred to a nursery; (4) the coffee plant is replanted in the field. After these activities, (5) cleaning and fertilizers/pesticides are applied to the plant; (6) the grain is harvested; (7) the process is finished by drying the grain under natural conditions (directly exposed to the sun); (8) the product is sold to the local organization. The materials involved are a hotbed, a nursery, shovels, and fertilizers/pesticides. All of this is for organic production during an average period of 4 years (see Figure 2).

### 3.2. Technology Routes of Indigenous Coffee Production

The following are the producers’ responses in relation to their expectations, current situation, and actions to be implemented (strategies). The priority expectations are as follows: (1) effective actions against coffee leaf rust; (2) soil analysis studies; (3) to have the resources for fertilizer development; (4) greater advice and accompaniment in the cultivation of coffee; and (5) studies of the interaction of the coffee plant with its environment. The current situation is as follows: (1) a high presence of coffee leaf rust that influences the low production of coffee; (2) problems with the nutrition of the soil; (3) low amount of resources (economic); (4) there is no proper pruning of coffee trees, control of new seedlings, or shade management practices; and (5) there are coffee plants without rust. For the above, the actions to be implemented are as follows: (1) monitoring and control of coffee leaf rust for 38.2% of coffee producers; (2) soil analysis for 26.5%; (3) creation of organic fertilizers for 17.6%; (4) accompaniment in the growth of the plant for 8.8%; and (5) experimentation of the interaction between coffee varieties and their environment for 8.8% (see Table 5, Figure 3).

### 3.3. Statistical Analysis

The evaluation of the data distribution, considering the sample size, was conducted using the Shapiro–Wilk W test, following [89]]. The result shows an abnormal distribution (*p* < 0.05). This result suggests the application of Spearman’s Rho as a non-parametric test (−1 < rş < 1), to measure the correlation of social variables, such as gender, age, time producing, and family members, and productive variables, such as production level, varieties, coffee tree age, cultivated area, flora, pruning, frequency, fertilizer, soil compost, type, additional crops, and pest incidence, with technology routes specifically for the actions to be implemented such as monitoring and controlling coffee leaf rust, soil analysis, the creation of organic fertilizers, accompaniment in the growth of the plant, and experimentation of the interaction between coffee varieties and their environment, based on the recommendations of [93].

For the social variables, the results show a positive correlation with gender (0.028) and a negative correlation with age (−0.009), time producing (−0.013), and family members (−0.063). 

The productive variables present a positive correlation with the production level (0.001), varieties (0.250), coffee tree age (0.162), pruning (0.312), pruning frequency (0.305), fertilizer (0.320), type of soil compost (0.121), additional crops (0.151), and pest incidence (0.041). However, negative results were found for the cultivated area (−0.112), flora (−0.058), and soil compost (−0.037) (see Table 6, Figure 4).

### 3.4. Technological Proposal

The positive statistical correlations between the social and productive variables and actions to be implemented, such as monitoring and control of coffee leaf rust, accompaniment during the growth of the coffee plant, soil analysis, the creation of organic fertilizers, and experimentation with the environment, provide the guideline to make the following technological proposal: The production level and pest incidence can be controlled by monitoring coffee leaf rust. For this, the development and involvement of producers through a mobile application for early detection of the disease is registered as a feasible option, promoting data management and processing to boost the learning process [41]. This can become a reality through alliances of producers with the Early Warning System for Coffee promoted by IICA [94], since it is a public virtual space [95], which is an application that evaluates pest incidence, providing answers for proper management, and has already been implemented in many countries.

This technology has been implemented in Guatemala and Nicaragua. In Guatemala, it has mainly been implemented in southern regions I, II, III, and IV [96], located 1000 to 2500 m.a.s.l. [97], with a population of over 8 million inhabitants between 15 and 59 years old, 42.9% of household men having a high school education [98], and 67.6% of people living in multidimensional poverty having health and food and nutritional security, education, decent employment, and access to services and housing [99]. In Nicaragua, Jalapa, located 679 m.a.s.l., has a total population of 67,148 inhabitants, of which 48.9% are older than 15 years, with an estimated 19% with primary and high school education [100], and an incidence of extreme poverty of 36.2% of its population [101]. Some of these indicators are similar to the persons involved in this study; therefore, the results suggest that this type of technology could be helpful. However, heterogeneity needs to be assessed to conclude on whether this technology can provide similar results [102].

In parallel, the introduction of this technology can be for accompaniment during the growth of the coffee plant. To influence the management of the level of production, proper management of the varieties at different ages, pruning development, and the reduction in the rust incidence should be implemented based on the existing correlation between the strategies and the production context. 

This proposal aligns with [103] by recommending that the sector be integrated into programs to monitor coffee diseases to improve decision making.

2.The type of soil compost is related to soil analysis. For this, sensors and computer tomography [104,105] can support measuring soil nutrients and provide images of the field properties in the coffee field. Invariably, a technological platform has to be used to collect data and develop recommendations for producers through an application on cell phones. From this, networked sensors and data processing are promoted for the learning process [41], to attend to the type of soil compost used for coffee, and for additional crops.

This technological strategy has been used in Colombia through sensors to measure the soil moisture and indicate variations [106]. At the same time, computer tomography has proved to be an essential digital tool for evaluating the physical–chemical properties of the soil in coffee-producing regions such as Colombia [107]. Producers can quickly adopt these tools because of their benefits and facilities [108].

In Colombia, coffee departments such as Magdalena, Cesar, and Cuca have multidimensional poverty of 19.6% to 33.4% [109]. Magdalena is located 5775 m.a.s.l., with 1,263,788 inhabitants; of these, 63.8% are 15 to 64 years old, with average schooling of 8.2 years [110,111,112]. Cesar is located 168 m.a.s.l., with 1,098,577 inhabitants; of these, 63.9% are between 15 and 64 years old, with an alphabetization level of 13.4%, and an average schooling of 8.1 years [111,113,114]. Cauca is located 1693 m.a.s.l., with 1,243,503 inhabitants; of these, 66.5% are between 15 and 64 years old, with a school level of 8.3 years [110,112,114].

This type of technology, related to soil analysis, could be developed from the knowledge and experiences of local institutions in order to be introduced into the coffee context as a soil laboratory (see [115]).

3.In relation to fertilizers/pesticides, the creation of organic fertilizers has a direct relationship. An option can be to reuse coffee pulp as a good source of compost. For this, a dome-shaped greenhouse to take advantage of solar radiation can be a simple technology since it does not require intervention for its management; it just needs the care of depositing and rotating the coffee pulp continuously to collect the black liquid that contains all the nutrients that it exudes after some time [116,117], which would improve work and safe activity within the sector.

A similar technological initiative has already been experienced in Vilcabamba, Ecuador, allowing comparisons of the efficiency and time of grain drying [118]. Vilcabamba is located 1700 m.a.s.l., with 4778 inhabitants; of these, 39.9% are between 30 and 64 years old, with a poverty level of 61.8%, and an illiteracy level of 5.09% [119,120].

4.Regarding varieties and their experimentation with the environment, due to the existence of coffee plants without rust, evaluation of the presence of phenolic compounds and proteins, following [121], is suggested. For this, spectrophotometric technology and sensors may be the best option to measure the presence of spores, aside from carbon, hydrogen, and oxygen, combined with the use of the technological platform to install in the context of the coffee plant.

Spectrophotometric and sensor technologies have already been considered in forage studies for cattle production systems in Veracruz, Mexico [122]. Therefore, these technologies can be standardized within the coffee sector (see Figure 5); however, these technologies generally require trained people, which could represent a disadvantage [123], but nowadays, these techniques have become easy to use, which increases the possibility of them being adopted by farmers [124].

## 4. Discussion

The number of indigenous people in the coffee-producing localities of El Pajarito, Llano Coyul, Ocotal, Guadalupe, and Buenavista in Oaxaca, Mexico, is proportionally similar to the 949 producers from 35 communities in Chiapas, recorded by [125]. The altitude at which the communities live, recorded as 1253, 960, 498, 1022, and 1325 m.a.s.l., is in line with [126], which reported that the ideal altitude for coffee production is between 1000 and 1300 m.a.s.l., or even a low altitude such as the coast. Similar findings apply to the origin of the producers [127], indicating that coffee farming is an activity on which many small Mexican indigenous producers depend. 

Regarding the academic level, the results obtained are a slightly lower than those reported by [128], establishing that most producers have a primary and secondary educational degree (9 years). This makes the reduction in the indigenous coffee-producing population in Mexico and the academic gap that technology can mitigate evident.

The access to computers (0% to 8.7%), cell phones (32.2% to 65.9%), and internet (1.05% to 59.09%) of indigenous communities from El Pajarito, Llano Coyul, Ocotal, Guadalupe, and Buenavista is similar to that reported in the same populations in Central and South American countries. For example, there is 2% computer access in Nicaragua and Colombia, 3% in Venezuela, 5% in Panama, and 8% in Ecuador, while there is 39% cell phone access in Chile, 48% in El Salvador, 53% in Panama, 54% in Ecuador, and 64% in Costa Rica, in similar communities. Regarding the internet, the situation is similar in Venezuela with 1% and Bolivia with 4%, while the condition is slightly better by 7% than Panama (highest), with 52% [129]. The extreme poverty results are similar to those reported by [130], pointing out the same situation in coffee producers in Colombia. The previous results show a technological gap between the indigenous communities in Mexico and Central and South America. However, actions must be collected and directed towards more specific strategies beyond justifying this situation due to the lack of investment and infrastructure, among others. The technological proposal of this study can be used for this purpose (see Table 7).

The gender of coffee producers from El Pajarito, Llano Coyul, Ocotal, Guadalupe, and Buenavista is similar to that found in [131], which reported that women’s participation is just 16% in Latin America, bowing to male involvement. Further, there were 1.6 more family members compared to this research, with an average of up of 5.2 members, while the average age of family members was 55 years old, which is 8 years younger in men and 3 years older in women compared to the ages reported in the present study. The negative correlation of this variable is in concordance with [132], maybe because the greater number of family members does not imply, necessarily, more dependents for the farmer [103]; in the future, it will be essential to know more details about family members. Concerning the time producing, the results are in line with [133], which registered 30 years and up to three generations of experience producing coffee in men, while for women, 15 years was registered. The results for the members per family are 0.2 less than those reported by [134], which registered 3.8 persons. This compassion shows the inclination towards the masculine gender in the agricultural sector, in addition to the advanced age of the producers, for which it is necessary to carry out concrete actions that could be supported by the new generations, taking advantage of their propensity for the adoption of technologies. The value of 0.28 in age aligns with [102], based on a meta-analysis of 77 studies related to technology adoption, concluding that as the age of farmers increases, the probability of technology adoption decreases. The same research concluded that the number of plots in the cultivated area has a negative impact on technology adoption; this is in concordance with the negative value obtained in this study. 

Regarding the coffee production level, the number of kilograms produced by inhabitants in the localities of El Pajarito, Llano Coyul, Ocotal, Guadalupe, and Buenavista was 404 kg less than that produced by their counterparts from Guerrero, Mexico, with 689 kg [59]. The coffee varieties used, namely, Catimor, Oro Azteca, Bourbon, and Geisha, are in line with those reported by [135], planted in Mexico. The coffee trees reported by [77] were 23 years older than the majority in this study (2–7 years 52.9%). Concerning the cultivated area, indigenous coffee producers own 0.27 ha more, on average, than those in the study of [136], with between 1 and 3 ha, while regarding the flora reported as part of the shade for coffee, the reports of Chalum and Cuil trees are similar to those of [137,138]. The development and frequency of pruning are in line with [23], which stated that it is essential to carry out pruning; however, these have been reduced. 

In relation to fertilizers/pesticides used against coffee leaf rust, the use of oxicloruro and organic products is in line with [139,140], finding these products to be effective solutions. In the application of soil compost, the results of [18] are inverse to those of the producers who do not fertilize since the recommendation is to apply it before planting the coffee tree and constantly during the first 3 years. This can be explained by the higher quantity of soil nutrients in the region studied. For the type of soil compost, the use of the organic type, compost, and coffee shells is in line with [141,142,143], highlighting the improvement. In the additional crops such as maize and bean, the results align with [144,145], except that coffee is less common. The price is 113 cents (US dollar) higher than that reported by [146] with 17,002 cents per pound, while the perception of the presence of coffee leaf rust is similar to that found in [147], where the incidence is perceived as high, and in [148], in the sense of a lack of attention to the health of the plant (see Table 8).

The coffee production process in the localities of El Pajarito, Llano Coyul, Ocotal, Guadalupe, and Buenavista is similar to that reported by [149], in relation to seed supply, preparation, and care of the nursery, cleaning, grain harvest, and natural or dry conditions in Chiapas, Mexico. 

The expectations and current situation expressed by indigenous coffee producers from El Pajarito, Llano Coyul, Ocotal, Guadalupe, and Buenavista are in line with those expressed by indigenous coffee producers from Guerrero, Mexico, in the sense of effective actions against coffee pests and diseases, due to the presence of coffee leaf rust and coffee berry borers. There are differences in the actions to be implemented; while the results of this study show the action of monitoring, their counterparts in Guerrero, Mexico, focused on knowledge and technology management networks, in addition to greenhouses [59].

The positive statistical correlations of the socio-productive context, specifically of age, production level, and additional crops, with actions to be implemented align with the results [61] in Chiapas, Mexico. These similarities create a coffee context predominantly led by men, where production and additional crops are two priority elements in which the application of actions to improve the sector is necessary. In this sense, the varieties’ positive correlation is in line with [150], mentioning their development and their combination with technological improvement as essential strategies for developing the coffee sector in Brazil. The age of coffee plantations and accompanying actions are in line with the empirical evidence related to implementing coffee tree renovation in Huila, Colombia [151]. This same accompaniment line is related to pruning, fertilizers, and soil compost by granting technological packages such as pruning, clearing, replanting, mineral nutrition with fertilizers, and soil improvers in Veracruz, Mexico [152]. Pest incidence and the relationship with control and monitoring are in line with [153], with the proposal to add strategies and actions to mitigate the incidence of rust in Latin America.

The proposal of a mobile application for the monitoring and control of rust is in line with the study by [154], which proposed a real-time mobile application for the early identification of various types of diseases in the coffee tree in India. This digital tool grants, *per se*, an accompaniment in the development of the coffee plant. The implementation of soil analysis through sensors and computers is in line with the proposal developed by [155], where the similarity in the technological guideline in South America is evident. 

The dome-shaped greenhouse proposal for creating organic fertilizers reusing the coffee pulp is in line with the document of [156], which recommends developing the process under a roof with a cement floor for turning the materials in the preparation of fertilizers. The recommendation for the use of spectrophotometric technology for the experimentation between coffee varieties and their environment is in line with [157], which used this technology for the analysis of coffee leaves in Veracruz, Mexico, which could be extrapolated for the presence/absence of rust spores and their environment (see Table 9).

## 5. Conclusions

Based on the objectives to explore coffee’s social, productive, and digital contexts, to determine technological routes, and to propose digital technologies, this study recommends the introduction of a mobile application to monitor and control coffee rust due to the level of production and the incidence of pests; sensors and computers for soil analysis; a dome-shaped greenhouse for the creation of organic fertilizers; and spectrophotometric technology due to the existence of coffee plants without rust, focusing on the varieties and their experimentation with their environment. These technologies can be introduced to the localities of El Pajarito, Llano Coyul, Ocotal, Guadalupe, and Buenavista since they have social, productive, and digital elements, besides being a requirement of the indigenous coffee-producing community in the area, and the fact that they have already been introduced in regions in Central and South America such as Guatemala, Nicaragua, Colombia, Ecuador, and Mexico, with similar social, productive, and digital characteristics. For this, Llano Coyul indigenous coffee producers have the best social and digital features, registering the highest rates of formal education, internet, and mobile cell phones; it is recommended to start introducing Industry 5.0 technologies in this locality, which will have a positive impact.

For the above, the hypothesis stated, i.e., “the technological proposal can improve the local development of the coffee tree due to its availability of resources and its application in a similar context based on the requirements of indigenous coffee producers”, is accepted.

The findings of this study attempt to contribute to the existing knowledge by (1) providing information about coffee’s social, productive, and digital contexts; (2) presenting technological routes based on the expectations, current situation, and actions that indigenous producers of coffee could implement; and (3) establishing a technological proposal for developing coffee plants centered on indigenous producers’ requirements in Oaxaca, Mexico. In addition, this research improves the current state of the art by being considered as a model for making a technological proposal. The methodology design has scientific rigor and social relevance to different agricultural areas in Mexico. 

The limits of this research relate to two aspects: (1) the distance, since the localities of this study are more than 300 km away from the state capital, Oaxaca; (2) the alliances with the producers. It is suggested to consider these points in future research.

However, the results should be approached with caution because more studies are necessary in the guideline of technologies taking the human being as the heart of the research; a deep analysis of the heterogeneity of variables following [97]] is required, besides implementing the present technological proposal in this context as the continuation of the current study. For this, it is necessary to have alliances with governments, politicians, and technology developers to help reduce the digital gap.

## Figures and Tables

**Figure 1 plants-11-01502-f001:**
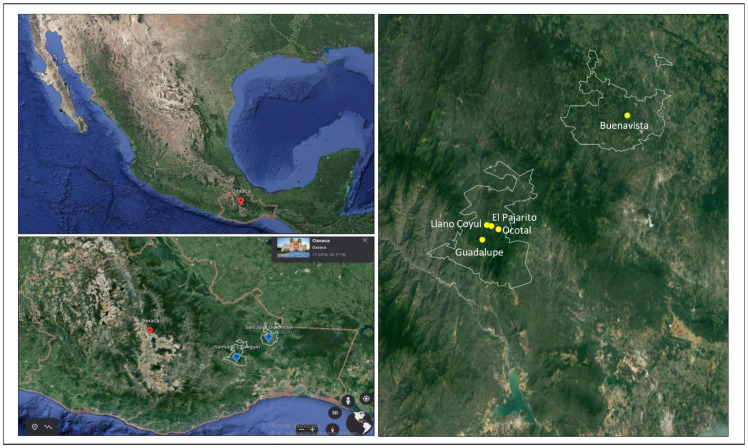
Location of El Pajarito, Llano Coyul, Ocotal, Guadalupe, and Buenavista in Santiago Lachiguiri and San Juan Guichicovi in Oaxaca, Mexico.

**Figure 2 plants-11-01502-f002:**
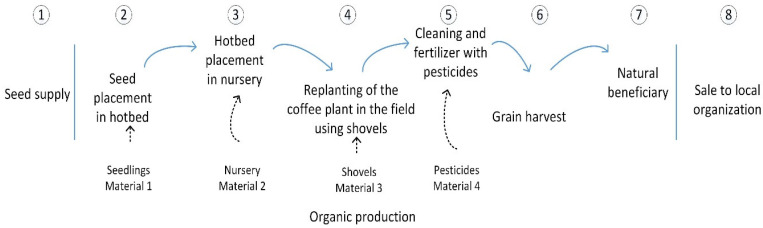
Process and materials of coffee production.

**Figure 3 plants-11-01502-f003:**
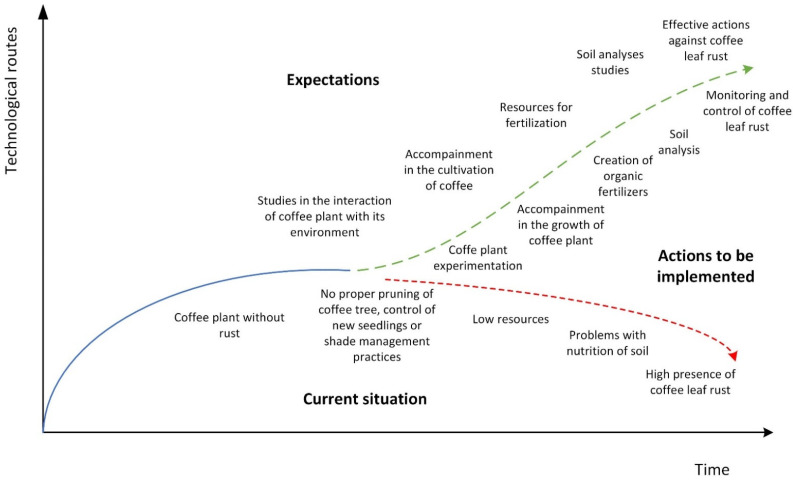
Technology routes.

**Figure 4 plants-11-01502-f004:**
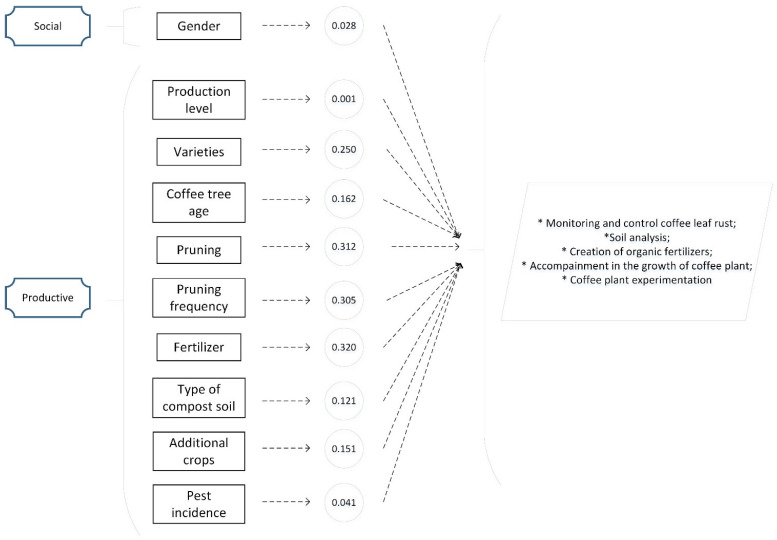
Positive statistical correlations.

**Figure 5 plants-11-01502-f005:**
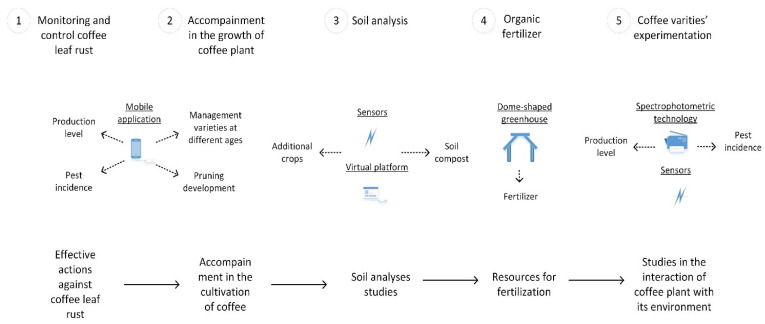
Technological proposal.

**Table 1 plants-11-01502-t001:** Technologies introduced with a human approach.

References	Human Approach, Social and Productive Context	Methodology for Technological Detection	Technological Proposal	Goal
[61]	Small coffee producers; an age of 40–44 years old; a cultivated area of 0.1–5 ha; a production level of 500–1000 kg p/ha.	(1) Technology roadmapping; (2) Digital compass; (3) Statistical analysis.	Statistical control process and digital management.	To detect emerging technologies for improving the sustainability of the coffee supply chain.
[62]	Mezcal producers; 44 years old; 27 years producing mezcal; a price of USD 14.6 per liter.	(1) Socialization, externalization, combination, and interiorization; (2) Evaluation and selection of technologies; (3) Statistical analysis.	Fiber optic refractometer, metal rooftop, horizontal distiller-fractionator, metal containers, digital platform, app and software for improving mezcal production.	To propose technologies for the sustainability of agave-mezcal production.
[63]	Milk and cheese producers; an age of 52 years old on average; secondary school education (mostly); a farm size of 5 ha; a herd number of 15.	(1) Farmer characteristics; (2) Use of technologies; (3) Interest of farmers of four groups.	(1) Use of smartphone (groups 1, 2, and 3); (2) Use of messages (groups 1 and 4); (3) Use of television and radio (group 1).	To make evident technological differences among farmers, and to design particular strategies.
[64]	Protected agriculture producers	(1) Competitiveness; (2) TIC assimilation; (3) Statistical analysis.	Assimilation and communication technologies.	To show the correlation between technology assimilation level and competitiveness.
[65]	Pumpkin seed producers; 10 years producing; cultivated area of 5 ha; cooperative members.	Open questions	Technification of sowing and harvesting, storage centers, and business plan.	Direct market articulation by production cycle.
[66]	Mexican vegetable producers	Theoretical revision.	Internet and apps to obtain information about weather, prices, pests, diseases, etc.	To propose that producers stay informed through cell phones; advice from extension services on internet portals; actions for assimilation of information technologies.
[67]	Vegetable, flower, and maize producers; age between 14 and 83 years old; a cultivated area of 0.01–8.0 ha.	(1) Open and closed questions; (2) Statistical analysis.	Agrochemicals.	To identify factors that influence the use of agrochemicals for developing productive reconversion and reducing the impact on health and the environment.

**Table 2 plants-11-01502-t002:** Social and digital contexts of El Pajarito, Llano Coyul, Ocotal, Guadalupe, and Buenavista in Santiago Lachiguiri and San Juna Guichicovi in Oaxaca, Mexico.

	Santiago Lachiguiri and San Juan Guichicovi				
Social and Digital Contexts/Locality	El Pajarito	Llano Coyul	Ocotal	Guadalupe	Buenavista
Total population	66	146	288	555	873
m.a.s.l.	1253	960	498	1022	1325
Formal education (years)	6.04 men	6.79 men	5.05 men	6.77 men	5.92 men
	5.36 women	5.03 women	4.36 women	5.28 women	4.36 women
Computer, laptop, or tablet (household)	0%	4.5%	3.3%	8.7%	2.4%
Cell phone (household)	52.6%	65.9%	32%	48.09%	56.1%
Internet access (household)	52.6%	59.09%	6.6%	29.5%	1.05%
Poverty	Extreme				
Language	Indigenous and Spanish				

**Table 3 plants-11-01502-t003:** Characteristics of indigenous coffee producers from El Pajarito, Llano Coyul, Ocotal, Guadalupe, and Buenavista in Santiago Lachiguiri and San Juan Guichicovi in Oaxaca, Mexico.

Indigenous Coffee Producers’ Characteristics	Men	Women
Gender	91.1%	8.8%
Age (average)	63	52
Time producing (years on average)	41.2	11.5
Family members	3.6	

**Table 4 plants-11-01502-t004:** Productive context.

Productive Context	Coffee Crop
Production level	285 kg/ha
Varieties	Catimor (47.2%), Oro Azteca (14.7%), Bourbon (14.7%), Geisha (11.7%), others (11.7%)
Coffee tree age	Less than two years (32.4%), two to seven years (52.9%), more than seven years (14.7%)
Cultivated area (average)	3.3 ha
Flora	Chalum (58.8%), Cuil (20.6%), other (20.6%)
Pruning coffee tree	No (61.7%), Yes (38.3%)
Pruning frequency	Every year (20.7%), two years (5.8%), three years (5.8%), four years (5.8%), do not prune it (61.9%)
Fertilizers/pesticides against rust	Oxicloruro (52.9%), organic (11.7%), others (14.8%), do not fertilizer it (20.6%)
Soil compost	No (20.6%), Yes (79.4%)
Type of soil compost	Organic (38.2%), compost (26.4%), coffee shells (14.7%), no soil compost (20.7%)
Additional crops	Nothing (47.1%), maize and bean (52.9%)
Sale price	USD 1.94
Pest incidence perception	Nothing (61.8%), little (26.5%), a lot (11.7%)

**Table 5 plants-11-01502-t005:** Technology roadmapping of coffee.

Technology Roadmapping	Indigenous Coffee Producers
Actions to be implemented	Monitoring and controlling coffee leaf rust (38.2%); soil analysis (26.5%); creation of organic fertilizers (17.6%); accompaniment in the growth of the plant (8.8%); coffee plant experimentation (8.8%).

**Table 6 plants-11-01502-t006:** Results of statistical analysis.

Social/Productive Variables	Statistical Correlations with Actions to be Implemented
Gender	0.028
Age	−0.009
Time producing	−0.013
Family members	−0.063
Production level	0.001
Varieties	0.250
Coffee tree age	0.162
Cultivated area	−0.112
Flora	−0.058
Pruning	0.312
Pruning frequency	0.305
Fertilizer	0.320
Soil compost	−0.037
Type of soil compost	0.121
Additional crops	0.151
Pest incidence	0.41

**Table 7 plants-11-01502-t007:** Discussion of variables of the context.

Variables	El Pajarito, Llano Coyul, Ocotal, Guadalupe, and Buenavista	Other Regions
Population	66 to 873 indigenous inhabitants from Oaxaca, Mexico	949 indigenous inhabitants from 35 communities in Chiapas, Mexico
m.a.s.l.	48 to 1325	1000 to 1300, low altitudes, and coast as ideal for coffee growth
Formal education (years)	5.05 to 6.79 for men 4.36 to 5.36 for women	9 years
Computer	0% to 8.7%	2% to 8% for indigenous regions in Nicaragua, Colombia, Venezuela Panama, and Ecuador
Cell phone	32.2% to 65.9%	39% to 64% for indigenous regions in Chile, El Salvador, Panama, Ecuador, and Costa Rica
Internet	1.05% to 59.09%	1% to 52% for indigenous regions in Venezuela, Bolivia, and Panama
Poverty	Extreme	Extreme in Colombia

**Table 8 plants-11-01502-t008:** Discussion of social and productive variables.

Variables	El Pajarito, Llano Coyul, Ocotal, Guadalupe, and Buenavista	Other Regions
Gender	91.1% men 8.8% women	84% men and 16% women in Latin America
Age	63 men 52 women	55 in Latin America
Time producing (years)	41.2 for men 11.5 for women	>30 for men and 15 for women in Copalita river basin in Oaxaca, Mexico
Family members	3.6 members	3.8 members in Puebla, Mexico
Production level	285 kg/ha	689 kg/ha in Guerrero, Mexico
Varieties	Catimor, Oro Azteca, Bourbon, and Geisha	Catimor, Oro Azteca, Bourbon, and Geisha in Mexico
Coffee tree age	2–7 years	>30 years in Jalisco, Mexico
Cultivated area	3.27 ha	3 ha in Hidalgo, Mexico
Flora	Chalum and Cuil	Chalum and Cuil in Chiapas and Oaxaca, Mexico
Pruning	Yes/No	Important, but has decreased
Fertilizers/pesticides against rust	Organic and oxicloruro	Organic and oxicloruro in Oaxaca, Mexico, and Guatemala
Soil compost	Yes/No	Applied before planting and during the first 3 years
Type of soil compost	9Organic, compost, and coffee shells	Organic, compost, and coffee shells in Mexico
Additional crops	Maize and bean	Maize and bean in Veracruz, Mexico
Sale price	USD 1.94/Kg	USD 1.7/pound
Pest incidence	Nothing, little, a lot	A lot in Guatemala

**Table 9 plants-11-01502-t009:** Discussion of process, routes, statistical analysis, and technological proposal.

Variables	El Pajarito, Llano Coyul, Ocotal, Guadalupe, and Buenavista	Other Regions
Process	Seed supply; seed placement in seedlings; seedlings placement in the nursery; replanting of coffee plant in the field; cleaning and fertilizer; grain harvest; natural conditions; sale to the local organization.	Seed supply, preparation, and care of the nursery; cleaning; grain harvest; natural conditions in Chiapas, Mexico.
Expectations; current situation; actions to be implemented (technological routes)	Effective actions against coffee leaf rust; high presence of coffee leaf rust; monitoring and control of coffee leaf rust.	Effective actions against coffee pests and diseases; presence of coffee leaf rust and coffee berry borers; knowledge and technology management networks and greenhouses.
Statistical analysis	Positive correlations between gender, production level, varieties, coffee tree age, pruning and frequency, fertilizer, type of soil compost, additional crops, and pest incidence and monitoring and control of coffee leaf rust, soil analysis, creation of organic fertilizers, accompaniment in the growth of the coffee plant, and coffee plant experimentation.	Positive correlations between gender, production level, and additional crops in Chiapas, Mexico; varieties and technology in Brazil; coffee tree age and accompaniment in Colombia; pruning, fertilizer, soil compost, and technological packages in Mexico; pest incidence and strategies and actions to mitigate the incidence of rust in Latin America.
Technological proposal	Mobile application for monitoring and controlling rust and accompaniment in the development of coffee plant; sensors and computer for soil analysis; a dome-shaped greenhouse for the creation of organic fertilizers; spectrophotometric technology for experimentation with the environment.	Real-time mobile application for identification of diseases in India; sensors and processors for soil analysis in South America; roof and cement floor for the preparation of fertilizers; spectrophotometric technology for the experimentation with varieties and their environment.

## Data Availability

The information used to support the findings are restricted by the National Council for Science and Technology of Mexico (Consejo Nacional de Ciencia y Tecnología CONACYT), according to the Federal Law on Transparency and Access to Public Information in Mexico. To know the access requirements, please contact the Technical Secretary of Transparency Committee, e-mail: ley_transparencia@conacyt.mx.

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
