# Peer review of "Roadmapping 5.0 Technologies in Agriculture: A Technological Proposal for Developing the Coffee Plant Centered on Indigenous Producers’ Requirements from Mexico, via Knowledge Management"

_plants, 2022, doi:10.3390/plants11111502_

Round 1

Reviewer 1 Report

The overall content is often unclear, and hard to follow. e.g.: line 62-63;  line 76-77; line 111: [51] together with students and fathers;  line 339; etc

The extensive introduction to the role of plants is irrelevant.

line 56-57: how significant are the affected areas? as a percentage of the World are with coffee prod?

How are the statistics on the social characteristics of coffee producers relevant to the investigated topic?

Conclusions: The ideas presented in the conclusion section are replicated from the results section and do not emphasize recommendations and the impact of the results on rural tourism. The authors should focus on pointing out what is their contribution to the existing knowledge and how their research improves the current state of art. The conclusion should be reformulated in elation to the argumentation in the paper.  Conclusions should also include a few words about the research limits, how the author intends to continue the research, and what direction to drive it.

Author Response

 May 25, 2022.

Cover letter: Responses to the reviewers’ comments

Dear

Prof. Dr. Jesus R. Millan-Almaraz and Dr. Luis M. Contreras-Medina.

Guest Editors.

Special Issue “Sensors and Information Technologies for Plant Development Monitoring”.

Plants.

Through this cover letter, we send the responses of the reviewers made to the manuscript entitled “Roadmapping 5.0 technologies in agriculture: A technological proposal for developing the coffee plant centered on indigenous producers’ requirements from Mexico, via knowledge management”. We appreciate the comments. All were valuable in reinforcing the manuscript. Thank you!

Responses to reviewer 1:

Comment 1: Extensive editing of English language and style required.

Response 1: Thank you for the comment. An exhaustive revision of the language and style in English was carried out. Please check the lines: 15, 16, 20, 36-38, 40, 44, 51-54, 56-59, 62-65, 67-69, 72-74, 79, 81-86, 88, 91-95, 97, 107-109, 111-112, 114-120, 122-124, 129, 132-133, 142, 161-163, 167-169, 172-173, 176-177, 182-183, 189-192, 195-198, 207, 211-214, 216, 218-219, 222-223, 227, 230-231, 235-239, 241, 247, 258-259, 268-269, 272, 277-278, 280, 282, 285-286, 289-290, 294-295, 298-299, 301-303, 316-318, 327, 334, 342, 632-366, 385, 395-396, 429-437, 451-453, 462, 485-486, 489, 501, 562-563, 565-566, 569-570, 582-585, 587-588, 592, 600-601, 604-608, 610, 621, 626, 633-634, 637-638, 640-641, 645, 656-659, 662-664, 666-667, 675, 677-681, 689-690, 693, 695, 698-700, 702-707, 710, 714-715, 718, 720-725, 727, 729, 738-739, 742, 745-746, 750, 752-753, 756, 758-760, 763-764, 766-767, 769, 774-777, 779, 789-804, 808-810, 816-817, 820, 822-850.

Comment 2: The overall content is often unclear, and hard to follow. e.g.: line 62-63; line 76-77; line 111: [51] together with students and fathers; line 339; etc.

Response 2: Thank you for the comment. The requested content was clarified, emphasizing the change from passive to active writing. Please check the lines in the manuscript: 67-68, 81-82, 114, 362, 365-366, 429-436, 451-453, 485-486, 559, 587-588, 657, 662, 667, 675, 698, 703, 724, 738, 742, 750, 758, 766, 774, 777, 808, 838.

Comment 3: The extensive introduction to the role of plants is irrelevant.

Response 3: Thank you for the comment. We tried to reduce the introduction; however, we believe that the information is essential to guide the reader step by step in the manuscript.

Comment 4: line 56-57: how significant are the affected areas? as a percentage of the World are with coffee prod?

Response 4: Thank you for the comment. It is estimated at 30% loss in Latin America as a percentage, representing 18.4% of the world. Please check the lines in the manuscript: 60-61.

Comment 5: How are the statistics on the social characteristics of coffee producers relevant to the investigated topic?

Response 5: Thank you for the comment. The role of social characteristics in the investigated topic was based on the assertion that is important for the digital transformation within the rural context. Please check the lines in the manuscript: 245-247.

Comment 6: Conclusions: The ideas presented in the conclusion section are replicated from the results section and do not emphasize recommendations and the impact of the results on rural tourism. The authors should focus on pointing out what is their contribution to the existing knowledge and how their research improves the current state of art. The conclusion should be reformulated in elation to the argumentation in the paper.  Conclusions should also include a few words about the research limits, how the author intends to continue the research, and what direction to drive it.

Response 6: The ideas of the conclusion were reformulated, emphasizing recommendations, the impact of results, the contribution of the existing knowledge, the provision to the state of the art, besides the limits, continuation, and direction in which the search must be driving in the future. Please check the lines in the manuscript: 789-792; 803-804; 808-810; 816-817; 820-849.

Reviewer 2 Report

The paper with the title “Roadmapping 5.0 technologies in agriculture: A technological proposal for developing the coffee plant centred on indigenous producers’ requirements from Mexico, via knowledge management”, in a complex study and technological proposal centred on indigenous small coffee producer requirements. Considering that coffee is a worldwide beloved beverage, this study is interesting for a very large audience from academics, economists and industry actors. The manuscript provides a detailed knowledge management analysis on the social, productive, and digital context of 5 localities from Mexico. The authors explored the relationship between various variables as the basis for an original proposal for an optimized technological approach.

The introduction is a bit long, I think it should be shortened a bit and the information presented more focused.

There are also many blank lines in the paper (eg lines 343-359; 380-401) the whole paper must be checked once again and the drafting errors corrected.

I consider this manuscript valuable for its original insights on coffee production sector and after the English revision of the text I consider it ready for publication.

Title - replace “centered” with “centred”

Manuscript – English syntax and grammar revision needed

Species name with lowercase letters, please change: Coffea Arabica to Coffea arabica; Hemileia Vastatrix to Hemileia vastatrix.

Author Response

 May 25, 2022.

Cover letter: Responses to the reviewers’ comments

Dear

Prof. Dr. Jesus R. Millan-Almaraz and Dr. Luis M. Contreras-Medina.

Guest Editors.

Special Issue “Sensors and Information Technologies for Plant Development Monitoring”.

Plants.

Through this cover letter, we send the responses of the reviewers made to the manuscript entitled “Roadmapping 5.0 technologies in agriculture: A technological proposal for developing the coffee plant centered on indigenous producers’ requirements from Mexico, via knowledge management”. We appreciate the comments. All were valuable in reinforcing the manuscript. Thank you!

Responses to reviewer 2:

Comment 1: The introduction is a bit long, I think it should be shortened a bit and the information presented more focused.

Response 1: Thank you for the comment. We tried to reduce the introduction; however, we believe that the information is essential to guide the reader step by step in the manuscript.

Comment 2: There are also many blank lines in the paper (eg lines 343-359; 380-401) the whole paper must be checked once again and the drafting errors corrected.

Response 2: Thank you for the comment. The blank lines were eliminated. Please check the lines: 144-156, 369-383; 405-425.

Comment 3: I consider this manuscript valuable for its original insights on coffee production sector and after the English revision of the text I consider it ready for publication.

Response 3: Thank you for the comment. A revision of the text was done. Please check the corrections in the lines: 15, 16, 20, 36-38, 40, 44, 51-54, 56-59, 62-65, 67-69, 72-74, 79, 81-86, 88, 91-95, 97, 107-109, 111-112, 114-120, 122-124, 129, 132-133, 142, 161-163, 167-169, 172-173, 176-177, 182-183, 189-192, 195-198, 207, 211-214, 216, 218-219, 222-223, 227, 230-231, 235-239, 241, 247, 258-259, 268-269, 272, 277-278, 280, 282, 285-286, 289-290, 294-295, 298-299, 301-303, 316-318, 327, 334, 342, 632-366, 385, 395-396, 429-437, 451-453, 462, 485-486, 489, 501, 562-563, 565-566, 569-570, 582-585, 587-588, 592, 600-601, 604-608, 610, 621, 626, 633-634, 637-638, 640-641, 645, 656-659, 662-664, 666-667, 675, 677-681, 689-690, 693, 695, 698-700, 702-707, 710, 714-715, 718, 720-725, 727, 729, 738-739, 742, 745-746, 750, 752-753, 756, 758-760, 763-764, 766-767, 769, 774-777, 779, 789-804, 808-810, 816-817, 820, 822-850.

Comment 4: Title - replace “centered” with “centred”.

Response 4: Thank you for the comment. The word centered was replaced by centred in the title, abstract. Please check the line in the manuscript: 3.

Comment 5: Manuscript – English syntax and grammar revision needed.  

Response 5: An exhaustive revision of the syntax and grammar was carried out. Please check the lines: 15, 16, 20, 36-38, 40, 44, 51-54, 56-59, 62-65, 67-69, 72-74, 79, 81-86, 88, 91-95, 97, 107-109, 111-112, 114-120, 122-124, 129, 132-133, 142, 161-163, 167-169, 172-173, 176-177, 182-183, 189-192, 195-198, 207, 211-214, 216, 218-219, 222-223, 227, 230-231, 235-239, 241, 247, 258-259, 268-269, 272, 277-278, 280, 282, 285-286, 289-290, 294-295, 298-299, 301-303, 316-318, 327, 334, 342, 632-366, 385, 395-396, 429-437, 451-453, 462, 485-486, 489, 501, 562-563, 565-566, 569-570, 582-585, 587-588, 592, 600-601, 604-608, 610, 621, 626, 633-634, 637-638, 640-641, 645, 656-659, 662-664, 666-667, 675, 677-681, 689-690, 693, 695, 698-700, 702-707, 710, 714-715, 718, 720-725, 727, 729, 738-739, 742, 745-746, 750, 752-753, 756, 758-760, 763-764, 766-767, 769, 774-777, 779, 789-804, 808-810, 816-817, 820, 822-850.

Comment 6: Species name with lowercase letters, please change: Coffea Arabica to Coffea arabica; Hemileia Vastatrix to Hemileia vastatrix.

Response 6: Thank you for the comment. The species names were changed with lower case letters. Please check the lines in the manuscript: 43, 206.

Reviewer 3 Report

Dear Authors, overall very nicely performed study however some suggestions for improvement are here below;

1) please specify the aim so it fits the main conclusions, for the reader to be clear

2) style and grammar, formatting check

3) visual improvement of Figures

4) suggest to cite in introduction part important sudies by Susilawati et al., e.g., "Coffee protects..." and "Coffee reduced the production...", which would significantly improve the scope of the multidisciplinary study

Author Response

 May 25, 2022.

Cover letter: Responses to the reviewers’ comments

Dear

Prof. Dr. Jesus R. Millan-Almaraz and Dr. Luis M. Contreras-Medina.

Guest Editors.

Special Issue “Sensors and Information Technologies for Plant Development Monitoring”.

Plants.

Through this cover letter, we send the responses of the reviewers made to the manuscript entitled “Roadmapping 5.0 technologies in agriculture: A technological proposal for developing the coffee plant centered on indigenous producers’ requirements from Mexico, via knowledge management”. We appreciate the comments. All were valuable in reinforcing the manuscript. Thank you!

Responses to reviewer 3:

Comment 1: please specify the aim so it fits the main conclusions, for the reader to be clear.

Response 1: Thank you for the comment. The objective was clarified in the conclusions, which were restructured for the reader’s clarity. Please check the lines in the manuscript: 789-792; 803-804; 808-810; 816-817; 820-849.

Comment 2: style and grammar, formatting check.

Response 2: Thank you for the comment. An exhaustive revision of the grammar and formatting English was carried out. Please check the lines: 15, 16, 20, 36-38, 40, 44, 51-54, 56-59, 62-65, 67-69, 72-74, 79, 81-86, 88, 91-95, 97, 107-109, 111-112, 114-120, 122-124, 129, 132-133, 142, 161-163, 167-169, 172-173, 176-177, 182-183, 189-192, 195-198, 207, 211-214, 216, 218-219, 222-223, 227, 230-231, 235-239, 241, 247, 258-259, 268-269, 272, 277-278, 280, 282, 285-286, 289-290, 294-295, 298-299, 301-303, 316-318, 327, 334, 342, 632-366, 385, 395-396, 429-437, 451-453, 462, 485-486, 489, 501, 562-563, 565-566, 569-570, 582-585, 587-588, 592, 600-601, 604-608, 610, 621, 626, 633-634, 637-638, 640-641, 645, 656-659, 662-664, 666-667, 675, 677-681, 689-690, 693, 695, 698-700, 702-707, 710, 714-715, 718, 720-725, 727, 729, 738-739, 742, 745-746, 750, 752-753, 756, 758-760, 763-764, 766-767, 769, 774-777, 779, 789-804, 808-810, 816-817, 820, 822-850.

Comment 3:  visual improvement of Figures.

Response 3: Thank you for the comment. The figures were improved to 500 dpi, in normal size. Please check the lines in the manuscript: 351, 446, 480, 526, 647.

Comment 4: suggest to cite in the introduction part important studies by Susilawati et al., e.g., "Coffee protects..." and "Coffee reduced the production...", which would significantly improve the scope of the multidisciplinary study.

Response 4: Thank you for the comment. The studies suggested were added to the introduction. Please check the lines in the manuscript: the third and fourth lines of the initial paragraph.

Round 2

Reviewer 3 Report

The topic is of outstanding interest and the changes made by authors throughout revision have risen the quaality significantly. Just minor spell and style to be done